# Numerical Study of Different Engineering Conditions on the Propulsive Performance of the Bionic Jellyfish Robot

Qiyun Cheng [1], Wenyuan Mo [2,*], Long Chen [2], Wei Ke [1], Jun Hu [2] and Yuwei Wu [2,*]

[1] Southern Power Grid Energy Development Research Institute Co., Guangzhou 510530, China
[2] School of Civil Engineering and Architecture, Hainan University, Haikou 570228, China
* Correspondence: redondomo@163.com (W.M.); yuweiwu@hainanu.edu.cn (Y.W.)

**Abstract:** Underwater robotics is rapidly evolving due to the increasing demand for marine resource exploitation. Compared with rigid robots propelled by propellers, bionic robots are stealthier and more maneuverable, such as autonomous underwater vehicles (AUVs) and remotely operated vehicles (ROVs), making them widely used underwater. In order to study the motion state of the umbrella jellyfish bionic robot, the displacement of the jellyfish robot along the same direction and the surrounding fluid pressure distribution caused by the jellyfish motion under different experimental conditions are discussed in this paper. The effect of different environmental factors on driving the jellyfish robot is determined by comparing the displacements at different observation points. The results of the study show that the lower the frequency and the longer the motion period, the greater the displacement produced by the robot within the same motion period. Frequency has a significant effect on the motion state of the jellyfish robot. While the change of amplitude also affects the motion state of the jellyfish robot, the displacement of the relaxation phase of the jellyfish robot is much smaller than that of the contraction phase with a small amplitude. It can be concluded that the effect of frequency on robot displacement is greater than the effect of amplitude on robot displacement. This study qualitatively discusses the changes of the motion state of the bionic jellyfish robot in still water under the excitation of different frequencies and amplitudes, and the results can provide corresponding reference for the future application of the bionic jellyfish robot, such as resource exploration, underwater exploration, and complex environment exploration.

**Keywords:** jellyfish; locomotion; swimming speed; ocean monitoring

## 1. Introduction

The oceans occupy 71% of the Earth's surface, but more than 95% of marine resources remain unexplored due to the complexity of the marine environment and the dangers of underwater operations, which in many cases prevent humans from conducting subsea operations [1,2]. The abundance of marine resources has stimulated the rapid development of underwater robots, in which the propulsion efficiency of autonomous underwater vehicles (AUVs) is particularly important. Compared with traditional rigid robots propelled by propellers, bionic robots are characterized by high environmental adaptability, high degrees of freedom, diverse functions, and continuous deformation [3–8]. In recent years, researchers have drawn inspiration from aquatic organisms to design a series of bionic robots with high mobility, low noise, high efficiency, and good concealment, making underwater robots have a broader application space [9,10]. Common bionic underwater robots include fish bionic [11], lobster bionic [12], tuna caudal fin bionic [13], turtle bionic [14], and jellyfish bionic [15–18]. Jet propulsion is used most widely in the propulsion systems of bionic robots [19]. Among them, common propulsion systems can be classified as the Bionic flapping wing propulsion system [20], Bionic water-jet propulsion system [21], Bionic wave-oscillating propulsion system [22], Body and/or caudal fin propulsion mode [23], and Median and/or paired fin propulsion mode [24]. Compared with other aquatic organisms,

jellyfish have a simple and flexible mode of motion and can adapt to more working environments, which can be used in marine environmental monitoring, energy exploration, fish tracking, ocean current observation, military reconnaissance, and weapons delivery [25]. Previous research has also made relevant analysis on the motion characteristics of the bionic jellyfish robot. Daniel [26], Dabiri et al. [27,28] Costello [29] Matthew and Jason [30] successively established the numerical model of the bionic jellyfish robot and analyzed the motion characteristics of jellyfish robots with different geometric shapes. Park [31,32] established the motion equation of the jellyfish in the contraction stage based on the material constitutive equation and studied the hydrodynamic characteristics of the bionic jellyfish robot based on this. However, the above research did not give a clear basis to improve the performance of the bionic jellyfish robot. Therefore, the purpose of this paper is to provide the basis for improving the motion efficiency of the bionic jellyfish robot in the still water environment by discussing the influence of different external force factors on the motion characteristics of the bionic jellyfish robot. Soliman, M simulated the 2D motion path of the turtle [33], while Andrew D. and others designed and verified through experiments that it is feasible to control the motion of a soft, highly compliant 2D manipulator [34]. In this study, in order to quickly model and efficiently analyze, the bionic jellyfish robot model was modeled in 2D in COMSOL (https://hal.archives-ouvertes.fr/hal02090402, accessed on 10 May 2022) Multiphysics 6.0 to observe its motion state under different environmental conditions [35]. In this paper, finite element modeling is performed by using COMSOL software for the water jet propulsion system of the bionic jellyfish to simulate the process of underwater motion of the bionic jellyfish robot in the still water environment. The effect of jellyfish motion under different amplitudes and frequencies are discussed. It has been found that the greater the amplitudes and the smaller the frequencies applied in a complete cycle, the greater the forward distance that can be achieved by the bionic jellyfish robot. The discussion and research on the motion characteristics of the bionic jellyfish robot and the bionic underwater robot will promote the exploration of underwater resources and the monitoring of the environment. This paper discusses the motion mode of the bionic jellyfish robot under different external forces, which provides a theoretical basis for the application of underwater resources development and environmental survey technology. It is worth noting that the simulation in this study is carried out under still water conditions. When simulating the motion state of the bionic jellyfish robot, the material properties and geometric characteristics of different joints and parts of the jellyfish robot are simplified. Therefore, the calculation results can only qualitatively discuss the motion characteristics of the jellyfish robot under different excitation conditions. The simulation closer to the actual project needs further study.

## 2. Methods and Model

The research object is the relationship between the motion state and excitation of the simplified bionic jellyfish robot in still water. In this study, a bionic jellyfish robot is assumed to be located in a homogeneous, incompressible fluid, and a turbulent model was not considered to save computational time. The Equations (1)–(3) describe the fluid control and momentum conservation in the $x$, and $y$ directions in the two-dimensional plane.

$$\frac{\partial u}{\partial x} + \frac{\partial w}{\partial y} = 0 \tag{1}$$

$$\frac{\partial u}{\partial t} + u\frac{\partial u}{\partial x} + w\frac{\partial u}{\partial y} = \upsilon\frac{\partial u^2}{\partial^2 x} + \upsilon\frac{\partial u^2}{\partial^2 y} - \frac{1}{\rho}\frac{\partial p}{\partial x} \tag{2}$$

$$\frac{\partial w}{\partial t} + u\frac{\partial w}{\partial x} + w\frac{\partial w}{\partial y} = -\frac{1}{\rho}\frac{\partial p}{\partial x} + \upsilon\frac{\partial w^2}{\partial^2 x} + \upsilon\frac{\partial w^2}{\partial^2 y} + g \tag{3}$$

where $u$ and $w$ are the velocities in $x$ and $y$ directions, respectively; $\rho$ is the fluid density; $p$ is the fluid pressure; and $g$ is the acceleration of gravity. In this model, the water density is $1000 \text{ kg/m}^3$ and dynamic viscosity is $1 \times 10^{-3}$ Pa·s.

This study specifies the horizontal rightward definitions as positive for $x$ and the vertical upward definitions for $y$ as positive. Similarly, the bending moment in the counterclockwise direction is specified as positive. As the bionic jellyfish robot moves through the fluid, it is subjected to both gravity and fluid forces. The bionic jellyfish robot model built in this study is polyethylene. Its density is $960 \, \text{kg/m}^3$, the Young's modulus is $3.5 \times 10^9$ Pa, and Poisson's ratio is 0.35. To simulate the interactions of water and jellyfish, as deformations of the solid was not considered in order to simplify the calculation, the equations of motion of the bionic jellyfish in the fluid may be expressed as follows.

$$m\frac{\mathrm{d}^2x}{\mathrm{d}t^2} = f_x \tag{4}$$

$$m\frac{\mathrm{d}^2y}{\mathrm{d}t^2} = f_y - mg \tag{5}$$

$$I\frac{\mathrm{d}^2\theta}{\mathrm{d}t^2} = M_{fg} \tag{6}$$

where $x$ and $y$ are the displacements of the solid in the $x$ and $y$ directions, respectively; $\theta$ is the angle of rotation of the solid; $f_x$ and $f_y$ are the components of the fluid force on the solid in the $x$ and $y$ directions; I is the rotational inertia of the solid around the center of gravity; $M_{fg}$ is the moment of the fluid force on the center of gravity.

In addition to having a flexible, flat, circular outer membrane (a gelatinous structure containing primarily water and extracellular proteins), jellyfish also have a ring of a muscle layer under their outer membrane, and eight natural muscle stimulators located at the outer edges of their bell-shaped outer membrane that stimulate different muscles independently. Jellyfish move in water by alternating between contractions and diastoles caused by rapid muscle contractions and slow muscle diastoles. Compared to other animals, the efficiency of this unique form of locomotion is remarkable. As shown in Figure 1, a simplified two-dimensional jellyfish model was developed to simulate the propulsion pattern of jellyfish in the water. There is a semicircular skeleton with a diameter of 50 mm on the head of the jellyfish. Tentacles of jellyfish are rigid pentagonal structures, and their tentacle joints are set as hinge connections without consideration of energy loss. Considering the influence of boundary on the jellyfish model swimming, the jellyfish robot model should avoid the influence of boundary effect on it as much as possible during the motion. The fluid domain is set as a square water with both length and width of 1 m. Among them, the two sides in the horizontal direction are set as wall conditions without slippage, and the ones on the upper and lower sides are set as open boundary conditions. Compared with other simulation software, such as Rock Gazebo [36] and freefloating gazebo [37], COMSOL has the advantage that it can add more relevant physical quantities to the problems studied, and react to more realistic engineering scenarios by coupling different physical fields. As shown in Figure 2, the bionic jellyfish robot model has been automatically meshed in COMSOL. Because there are many curve boundaries in the geometric model of the bionic jellyfish robot, in order to make the divided mesh better fit the boundary, the three node mesh method is selected to divide the geometric model in this study. There are 13,794 triangular mesh elements in the divided model. In order to observe the relationship between the displacement of different parts of the bionic jellyfish robot and time, the observation points are set at the head, tentacles, and joints as point 1, point 2, point 3 and point 4. The spatiotemporal distribution characteristics of the physical quantities of the motion characteristics of the bionic jellyfish robot under different excitation conditions were observed, respectively.

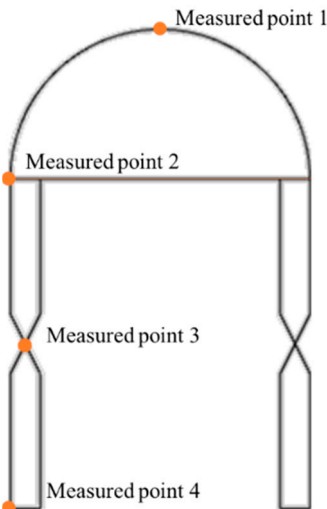

**Figure 1.** Schematic of jellyfish robot structure.

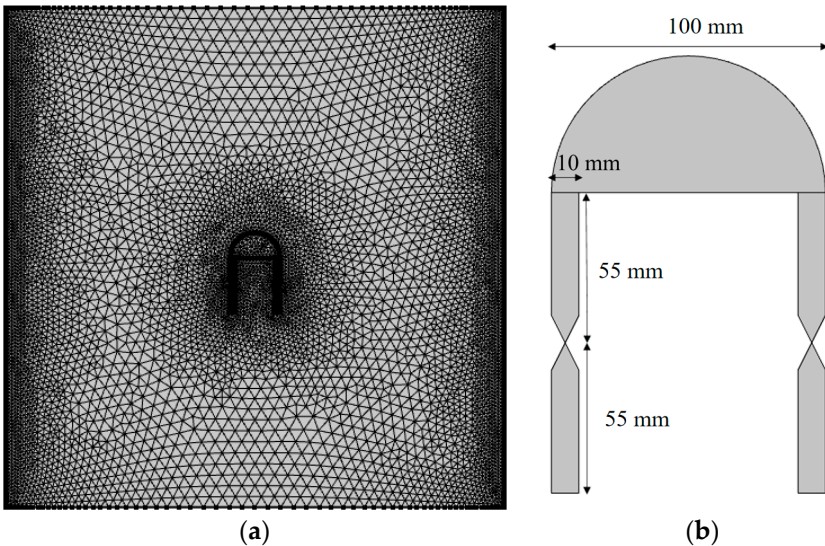

(**a**)  (**b**)

**Figure 2.** Schematic of model geometry and meshing. (**a**) Triangular finite-element mesh. (**b**) Geometric Model.

In this model, the antennae of the bionic jellyfish robot are connected by multiple links [38,39], external forces are applied at the joints, and the robot is propelled forward by the fluid motion induced by the tentacle rotation.

The bionic jellyfish robot is propelled forward by fluid motion created by rotating the tentacles in this model, with the torque applied as an external force to its tentacle joints. The jellyfish tentacle oscillation velocity is assumed to be sinusoidal in order to satisfy the initial velocity of the jellyfish robot motion and to allow it to oscillate normally in subsequent cycles. Since the mechanism of jellyfish swimming is that its velocity during contraction is greater than its velocity during diastole, it is assumed in this paper that the velocity at the tentacle joint during contraction is twice as fast as the velocity at the tentacle joint during relaxation. At the joint, the velocity period function of the bionic jellyfish robot is expressed as follows.

$$s(t) = \begin{cases} \sin(t) \ 0 < t \leq \pi/2 \\ \sin\left(2t - \frac{\pi}{2}\right) \pi/2 < t \leq \pi \\ \sin\left(t + \frac{\pi}{2}\right) \pi < t \leq 3\pi/2 \end{cases} \tag{7}$$

By deriving the velocity period function at this point, the acceleration function can be obtained. As torque is proportional to acceleration, the torque period function at the joint can be expressed as follows.

$$M(t) = \begin{cases} \cos(t) & 0 < t \le \pi/2 \\ 2\cos\left(2t - \frac{\pi}{2}\right) & \pi/2 < t \le \pi \\ \cos\left(t + \frac{\pi}{2}\right) & \pi < t \le 3\pi/2 \end{cases} \tag{8}$$

## 3. Analysis

The displacement of the measurement points of the bionic jellyfish robot is evaluated by varying the environmental parameters in the model. This set of controllable parameters includes the frequency of the torque period function and the magnitude of the amplitude. The driving effect of different amplitudes and frequencies on the bionic jellyfish robot was evaluated by comparing the pressure distribution and the distance traveled by the bionic jellyfish robot in one complete cycle. By observing the displacement of the bionic jellyfish robot under the excitation of different amplitude and period, the influence of this factor on the propulsion effect of the bionic jellyfish robot can be intuitively evaluated. By comparing the distribution of liquid pressure around the bionic jellyfish robot under different amplitude and period, the potential energy distribution of different parts can be observed, which has an important reference role in improving the appearance of the bionic jellyfish robot. Figure 3 shows a plot of the torque applied on the tentacle joints of the bionic jellyfish robot as a function of time under different experimental conditions. Among them, *M* is the external force, the torque, applied on the leg joints of the bionic jellyfish robot. *T* is the period of torque, and *F* is the amplitude of torque.

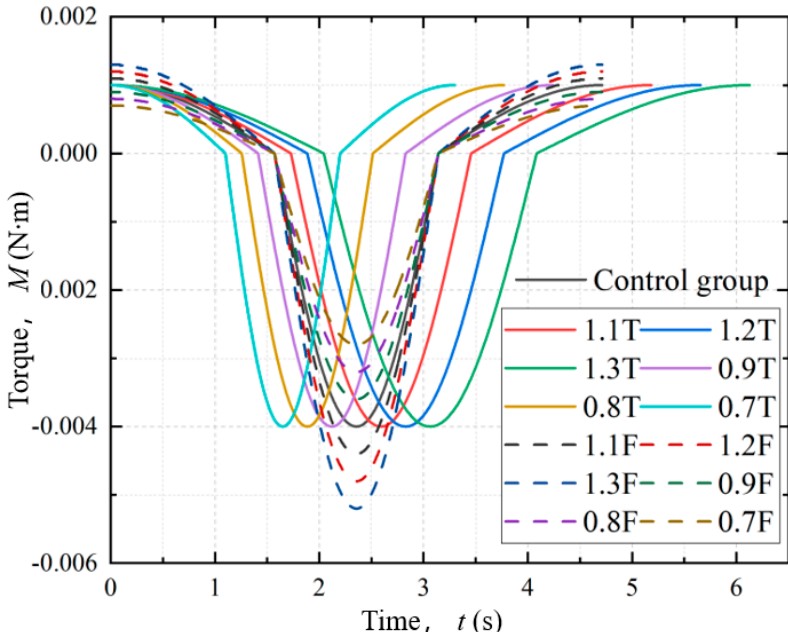

**Figure 3.** Applied torque versus time.

### 3.1. Frequency

The frequency of the external force will have a great impact on the movement of the bionic jellyfish robot. This section investigates the effect of frequency on stimulating jellyfish robot motion by comparing the displacement of the simulated jellyfish robot and the stress distribution in the diastolic and contraction states when the frequency of the applied torque is increased and decreased by 10%, 20%, and 30%, respectively, from that of the control group.

Figure 4 shows the displacement of the jellyfish robot in one cycle with different frequency stimuli. From Figure 4, it can be observed that the motion states of observation point 1 and observation point 2 on the head of the bionic jellyfish robot have similar trends, and the motion states of observation point 3 and observation point 4 on the tentacles are similar. In the same cycle, the lower the frequency and the longer the cycle time, the farther the displacement distance of the jellyfish robot. Conversely, when the frequency of torque is high and the cycle time is short, the smaller the displacement produced by the jellyfish robot. Observing Figure 4a,b, it can be seen that the effect of different frequencies on the head displacement during the diastolic and contraction stresses of the jellyfish robot does not vary much. Observing Figure 4c,d, it can be seen that different frequencies of torque have a very significant effect on the displacement of the tentacles in the contracted and diastolic states.

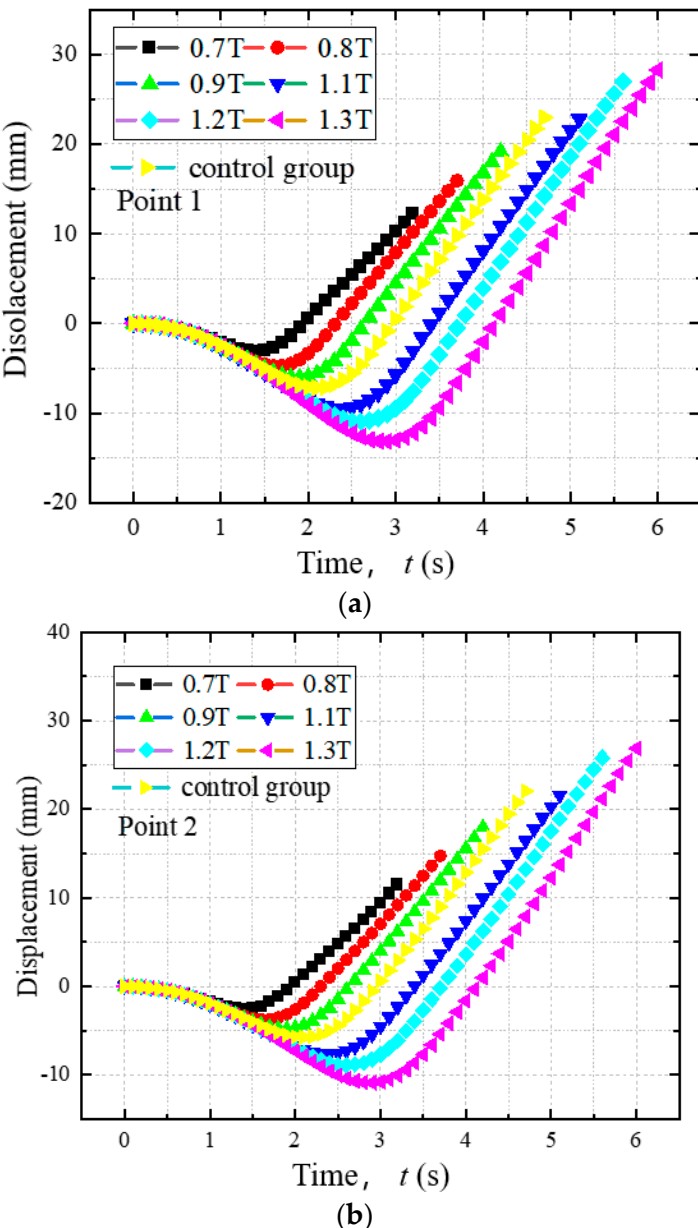

**Figure 4.** *Cont.*

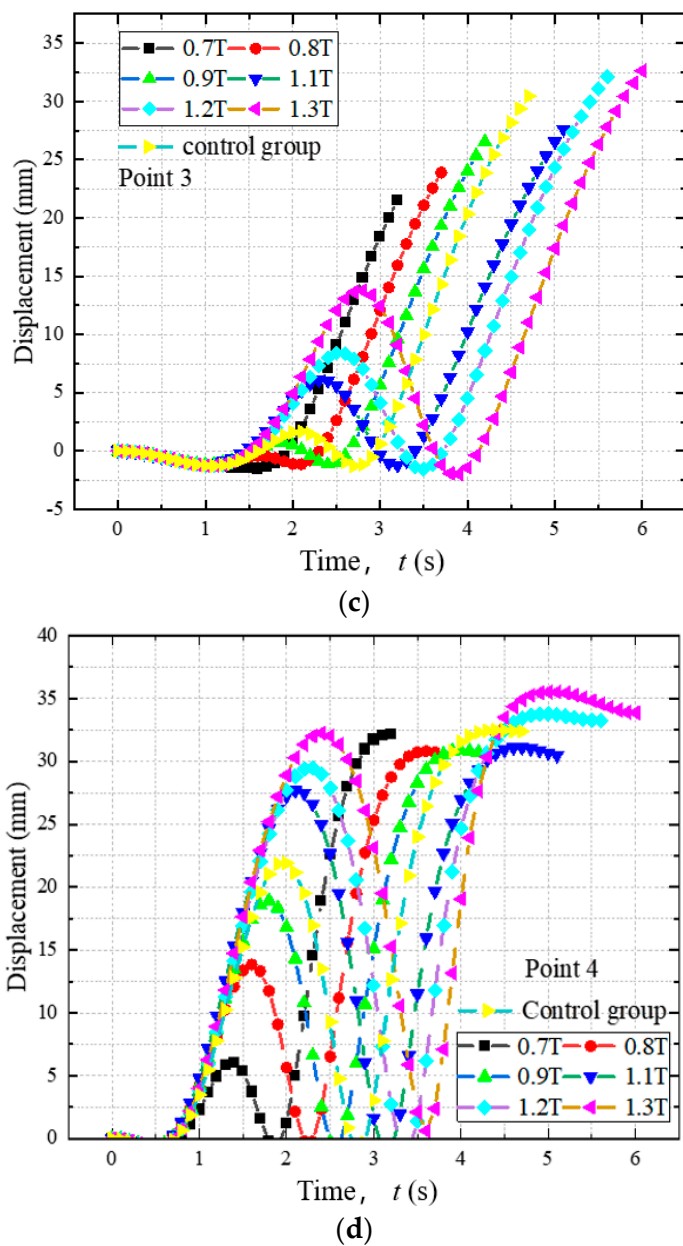

**Figure 4.** Displacement of different measurement points of the jellyfish robot under the action of different frequencies. (**a**) Point 1. (**b**) Point 2. (**c**) Point 3. (**d**) Point 4.

Figure 5 shows the pressure distribution of the surrounding fluid in the relaxed states of the bionic jellyfish robot under the action of different frequencies. As can be seen from Figure 5, the relaxation state of the bionic jellyfish robot exhibits significant differences at different frequencies of action. At low frequencies with long periods, the tentacles of the jellyfish robot folded more, and the pressure generated around the tentacles did not differ significantly at different frequencies. This is due to the fact that the longer period at low frequencies allows the robot to have more time to fold the tentacles in the diastolic state. As a result, the lower the frequency, the more fluid the jellyfish robot draws into the cavity.

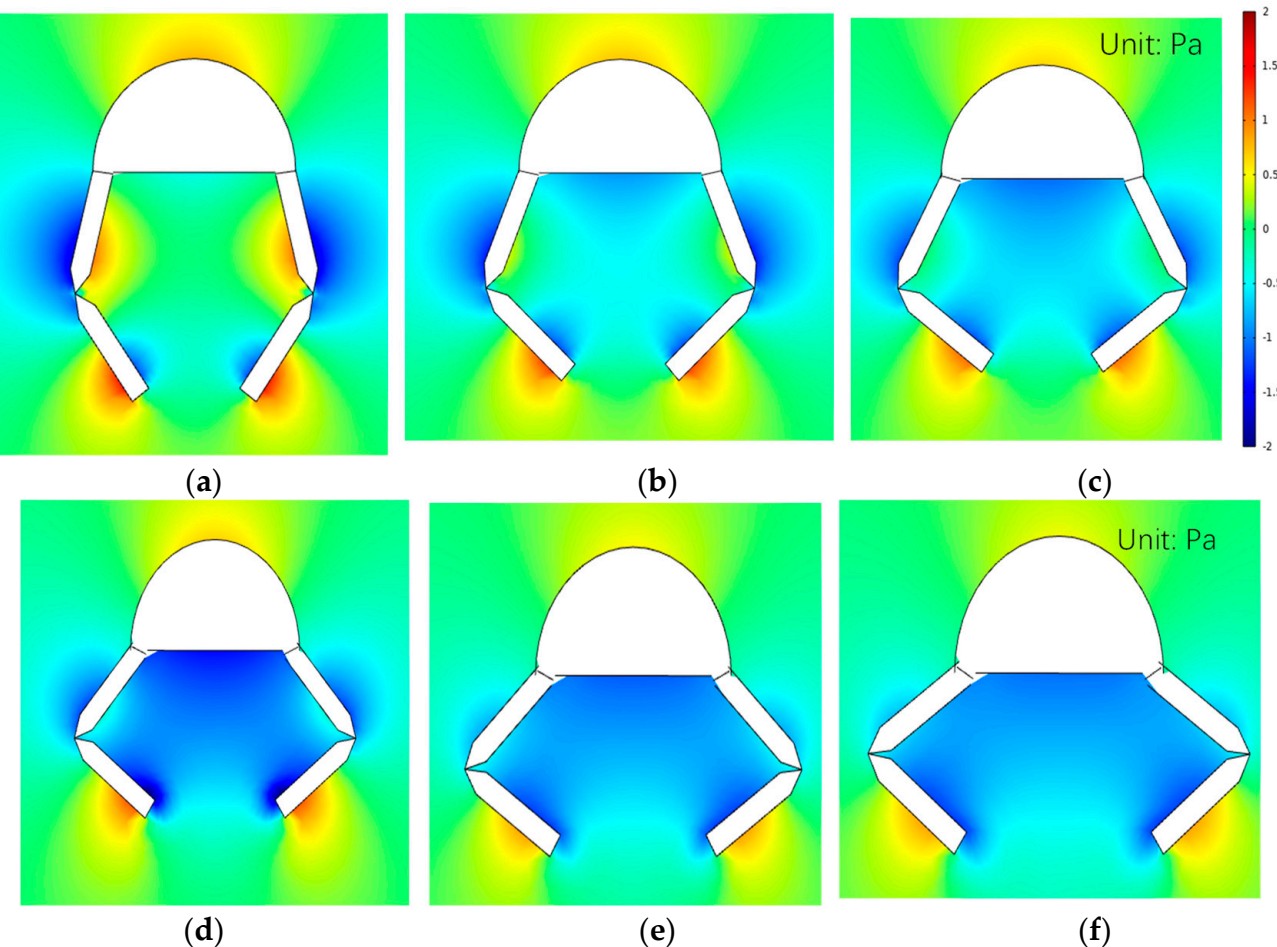

**Figure 5.** Fluid pressure distribution around the relaxed state of the jellyfish robot under the action of different frequencies. (**a**) 0.7 T. (**b**) 0.8 T. (**c**) 0.9 T. (**d**) 1.1 T. (**e**) 1.2 T. (**f**) 1.3 T.

Figure 6 shows the jellyfish robot in the contracted state under the action of different frequencies. As can be seen from Figure 6, the contraction state of the jellyfish robot is different for different frequency effects, and the tentacles of the jellyfish robot fold more when the frequency is higher and the period is shorter. This is due to the fact that the tentacles are easily turned when a torque in the opposite direction is applied to the tentacles since no more fluid is drawn into the cavity of the jellyfish robot during the diastolic phase.

*3.2. Amplitudes*

The amplitude of the excitation applied to the joint of the bionic jellyfish robot is also an important factor affecting the motion of the bionic jellyfish robot. In order to study the influence of amplitude on the driving of biomimetic jellyfish robot, the displacement and pressure distribution at different observation points of biomimetic jellyfish robot are discussed when the amplitude is increased or decreased by 10%, 20%, and 30%, respectively. Figure 7 shows the variation of the displacement of the bionic jellyfish robot at different observation points with time for different amplitude effects.

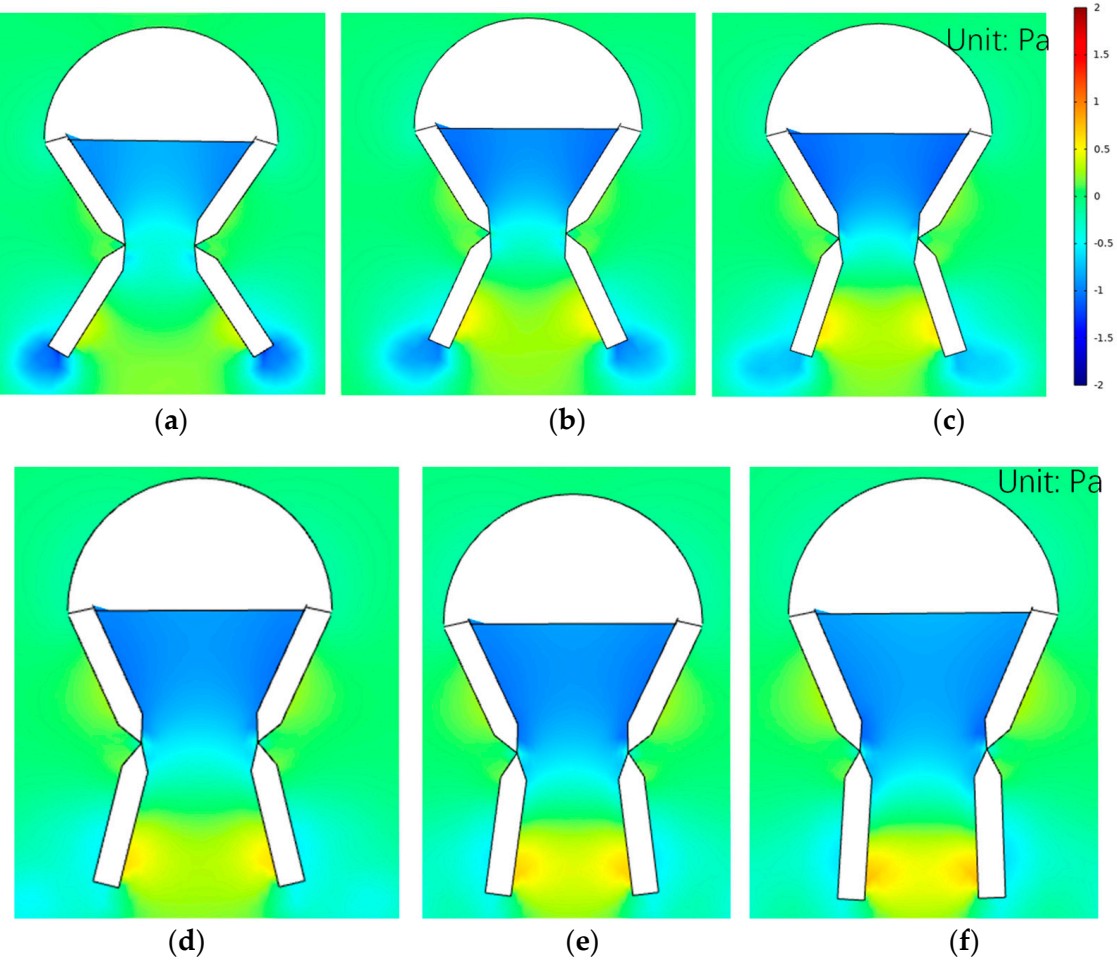

**Figure 6.** Fluid pressure distribution around the contracted state of the jellyfish robot under the action of different frequencies. (**a**) 0.7 T. (**b**) 0.8 T. (**c**) 0.9 T. (**d**) 1.1 T. (**e**) 1.2 T. (**f**) 1.3 T.

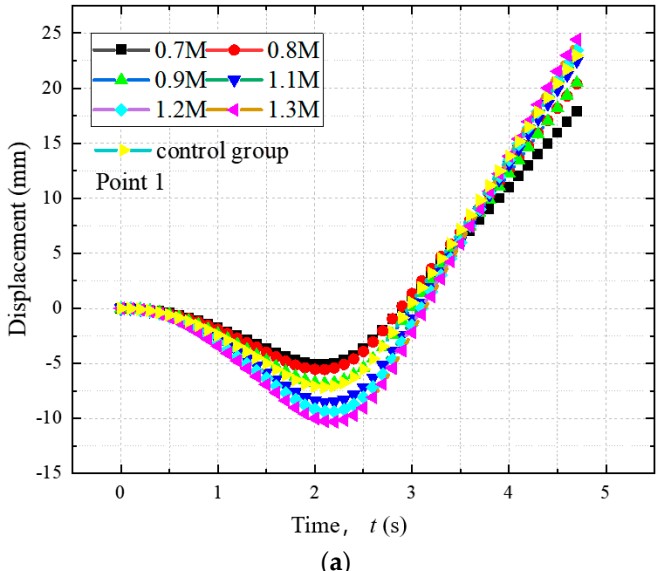

(**a**)

**Figure 7.** *Cont.*

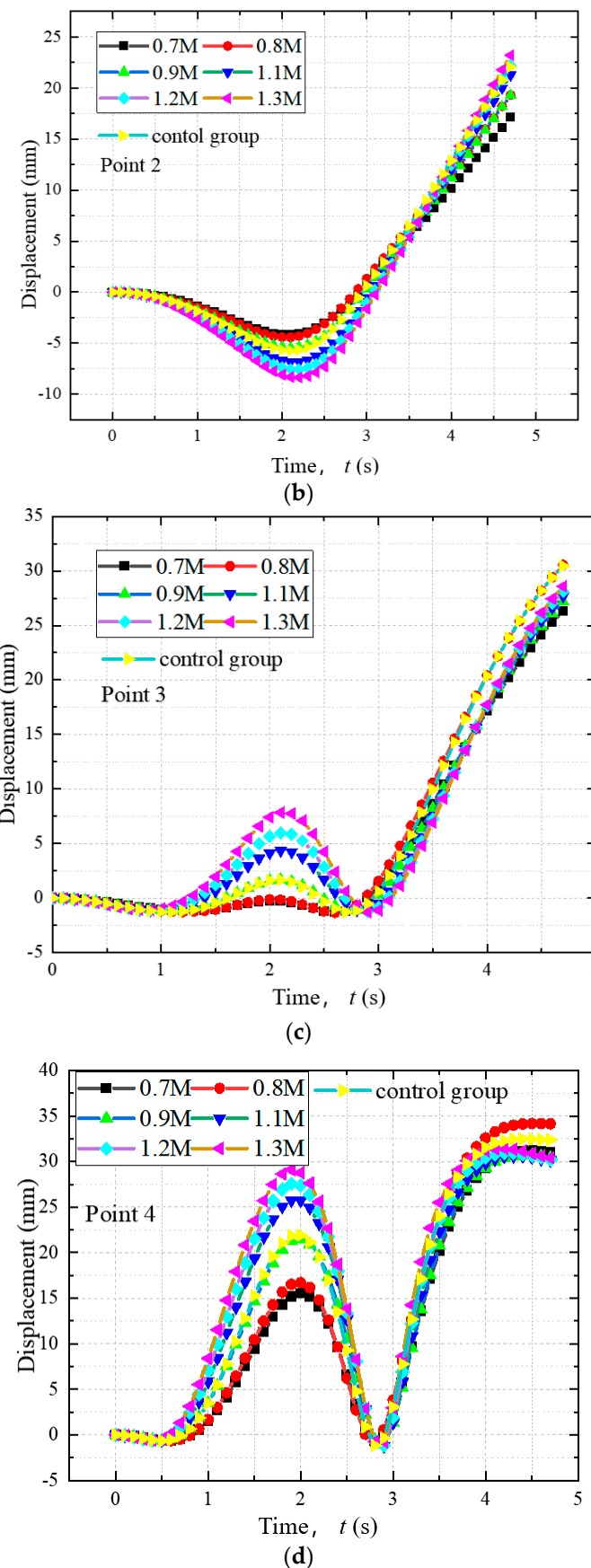

**Figure 7.** Displacement of different measurement points of the jellyfish robot under the action of different amplitudes. (**a**) Point 1. (**b**) Point 2. (**c**) Point 3. (**d**) Point 4.

Observing Figure 7, it can be observed that the effect of amplitude on the driving force of the bionic jellyfish robot is relatively small compared to the frequency. However, it can still be seen from the figure that the displacement at each observation point of the bionic jellyfish robot in the contraction phase increases significantly with the increase in amplitude. In the diastolic state, the displacement at different observation points varies, with the displacement at the head observation point increasing with the amplitude, and the displacement at the tentacle end observation point decreasing with the amplitude. The final displacement results at observation point 4 did not differ much for different amplitudes. This may be due to the fact that more work is required to change the direction of the contactor motion after applying a larger force to the joint with the same frequency and then applying a reverse force. Therefore, a smaller torque can produce a larger displacement in the contracted state.

Figure 8 shows the water pressure distribution in the water around the bionic jellyfish robot in the relaxiation state under the effect of different amplitudes. It can be seen that under the action of different amplitudes, the fluid pressure distribution around the relaxed bionic jellyfish robot shows significant differences. The larger the amplitude, the greater the fluid pressure around the bionic jellyfish robot in the relaxed state. This is because the larger amplitude allows the lower limbs of the bionic jellyfish robot to inhale more fluid during the movement process, so it will have greater potential energy and stress.

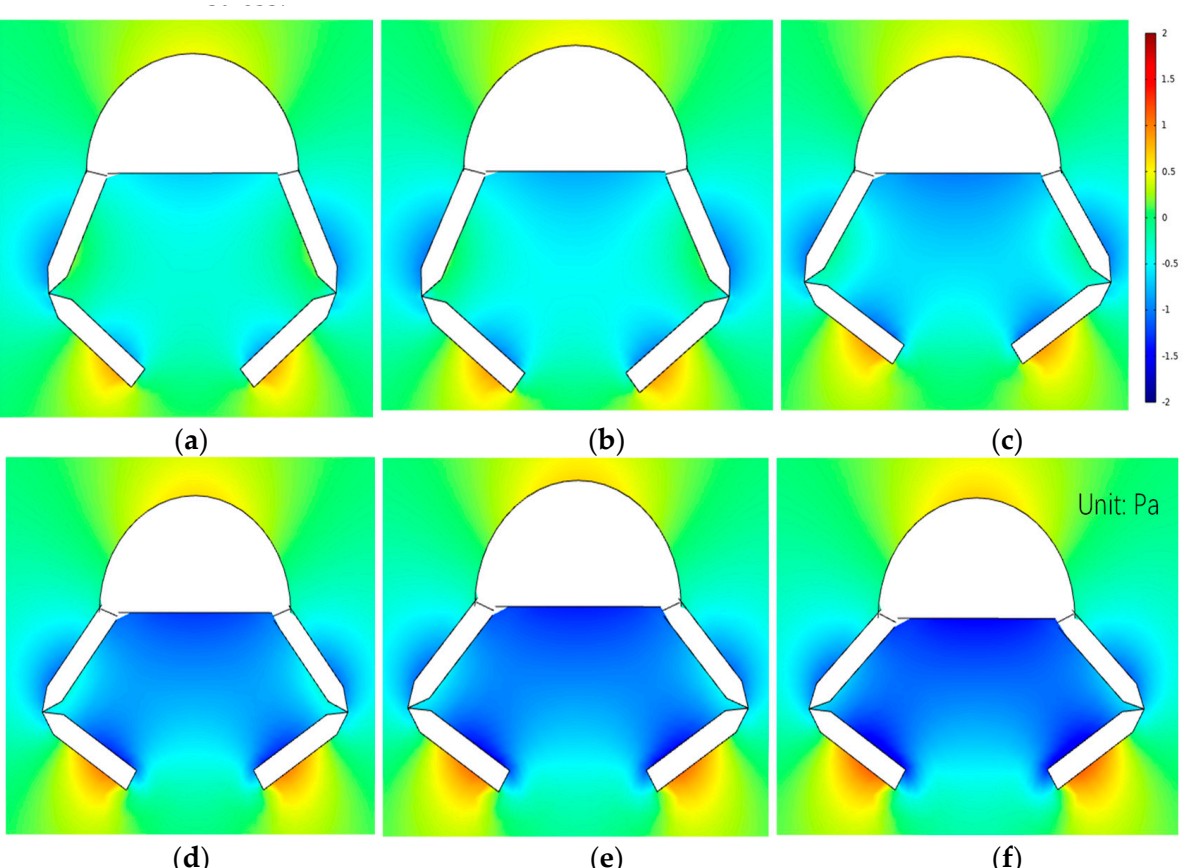

**Figure 8.** Fluid pressure distribution around the relaxed state of the jellyfish robot under the action of different amplitudes. (**a**) 0.7 M. (**b**) 0.8 M. (**c**) 0.9 M. (**d**) 1.1 M. (**e**) 1.2 T. (**f**) 1.3 M.

Figure 9 shows the fluid pressure distribution around the bionic jellyfish robot in the contracted state under different amplitudes. It can be seen that the fluid pressure distribution around the jellyfish robot increases with the increase in amplitude for different amplitudes. This is because a larger amplitude will produce a larger impact on the fluid in the cavity of the jellyfish robot, resulting in higher stress and potential energy.

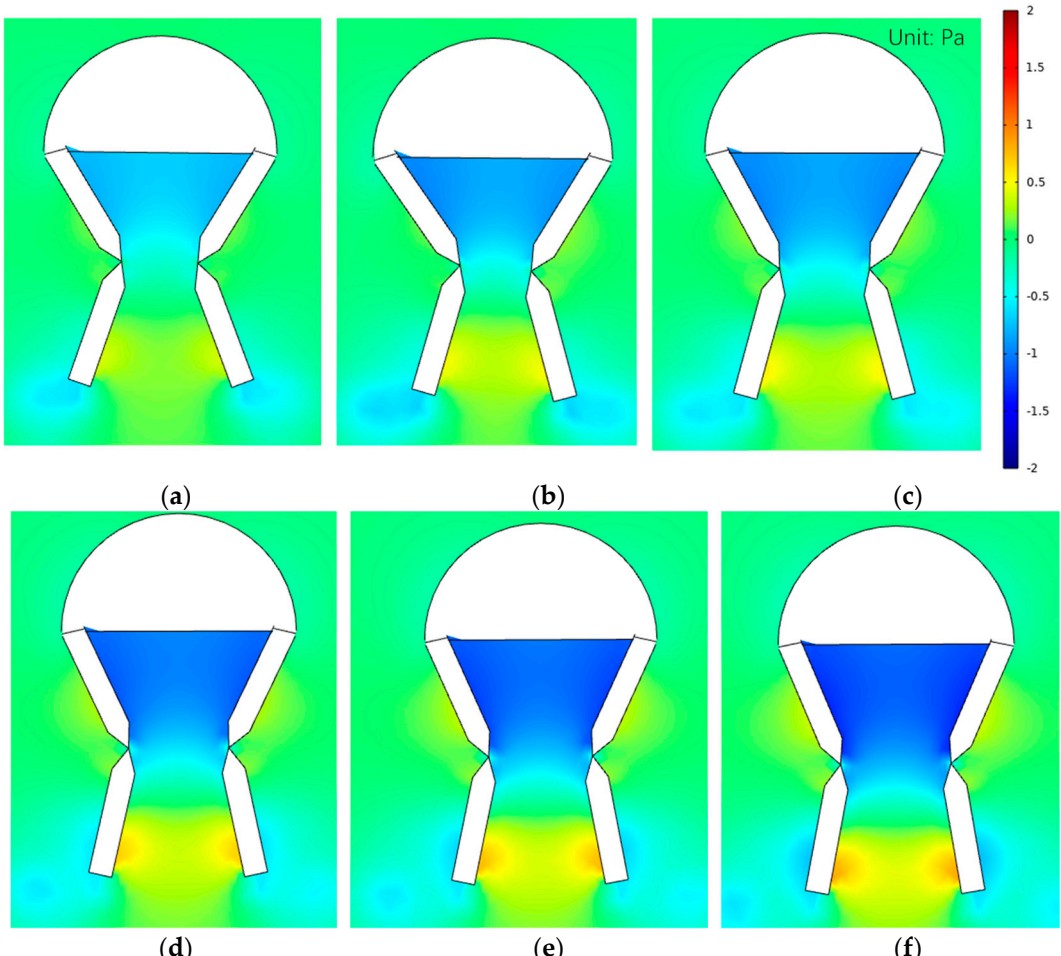

**Figure 9.** Fluid pressure distribution around the contracted state of the jellyfish robot under the action of different amplitudes. (**a**) 0.7 M. (**b**) 0.8 M. (**c**) 0.9 M. (**d**) 1.1 M. (**e**) 1.2 T. (**f**) 1.3 M.

In this section, the effects of different frequencies and amplitudes on the driving effect of the bionic jellyfish robot and the effect on the pressure of the surrounding fluid environment are discussed. It is worth noting that the testing environment of the bionic jellyfish robot in this study is still water. The simulation material of the bionic jellyfish robot is also simplified to uniform material. Therefore, the research on the motion state of the bionic jellyfish robot under the action of different frequencies and amplitudes is only qualitative. The quantitative study of fluid environment and motion trajectory still needs further exploration.

## 4. Conclusions

Based on the motion characteristics of jellyfish, the motion characteristics are observed, modeled, and analyzed for the forces at its tentacle joints. In this paper, by discussing the analysis of jellyfish robot motion states under the action of different amplitudes and frequencies in the still water environment, the following conclusions can be drawn.

1   The simplified bionic jellyfish robot model proposed in this paper can simulate its movement process in still water.
2   The effect of amplitude on the motion state of the bionic jellyfish robot is small, and the forward distance of the bionic jellyfish robot does not differ much under the action of different amplitudes.
3   The lower the frequency, the stronger the driving effect of the bionic jellyfish robot. In the same period range, the bionic jellyfish with lower frequency can advance a longer distance.

4     Frequency and amplitude have little effect on the pressure distribution of the fluid around the bionic jellyfish robot, but have a higher correlation on the motion pattern of the bionic jellyfish robot. When the frequency is lower, the jellyfish robot sucks less water into the cavity in the relaxed state.

In future work, the research will focus on quantifying the effect of fluid conditions, forms of robots, and driving forces on the movement performance. At the same time, analysis of the motion trajectory of the bionic jellyfish robot will also be taken into consideration. The developed bionic jellyfish robots can be used in the future for resource exploration and underwater exploration in unknown and complex environments.

**Author Contributions:** Conceptualization, W.K. and Q.C.; methodology, W.M.; software, W.M.; validation, Y.W., W.M. and L.C.; formal analysis, W.K.; investigation, W.K.; resources, W.K.; data curation, Y.W.; writing—original draft preparation, Y.W.; writing—review and editing, Y.W.; visualization, W.K.; supervision, W.K.; project administration, L.C.; funding acquisition, W.K. and J.H. All authors have read and agreed to the published version of the manuscript.

**Funding:** This research was funded by the Natural Science Foundation of Hainan Province (No. 519MS024), Natural Science Foundation Innovative Research Team Project of Hainan Province, China (522CXTD511). The authors also acknowledge support from the China Scholarship Council (201907565040).

**Institutional Review Board Statement:** Not applicable.

**Informed Consent Statement:** Not applicable.

**Data Availability Statement:** Not applicable.

**Acknowledgments:** This research was supported, in part, by the Natural Science Foundation of Hainan Province (No. 519MS024) The authors also acknowledge support from the China Scholarship Council (201907565040).

**Conflicts of Interest:** The authors declare no conflict of interest.

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
