# Peer review of "Numerical Study of Different Engineering Conditions on the Propulsive Performance of the Bionic Jellyfish Robot"

_sustainability, doi:10.3390/su15054186_

Round 1
Reviewer 1 Report (Previous Reviewer 1)

Author Response
Please see the attachment

Reviewer 2 Report (Previous Reviewer 4)
2022/12/22 (Sustainability)
Review Comments for Manuscript Number: sustainability-2130100-peer-review-v1
|
Title: |
Numerical Study of Different Engineering Conditions on the Propulsive Performance of the Bionic Jellyfish Robot |
|
Journal: |
Sustainability |
The authors have provided all the required revisions in the revised manuscripts. Besides that, they answered all the questions. However, I still believe that all the parameters should be presented in this study in order to be in complete shape.
Author Response
Please see the attachment

Reviewer 3 Report (Previous Reviewer 5)
The authors answered all questions and comments. The manuscript can be published.
Author Response
Please see the attachment

Reviewer 4 Report (Previous Reviewer 6)
The quality of the manuscript has been improved, and the authors have addressed the reviewer's questions properly.
Round 2
Reviewer 1 Report (Previous Reviewer 1)
All the questions and problems have been well addressed. The quality of this paper has been improved.
This manuscript is a resubmission of an earlier submission. The following is a list of the peer review reports and author responses from that submission.
Round 1
Reviewer 2 Report
The problem with the paper is that it is only a simulation and they don't consider plenty of issues that happen in real life.
The paper only considers only one challenge of the robot they would like to develop.
There must be a prototype.
It is absolutely necessary to test it in amock-up.
Reviewer 3 Report
Please include in your analysis the following factors:
1- Effect of the depth and viscousity of liquid on the structure integrity.
2- Effect of the depth and viscousity of liquid on the Navigation speed and displacement in both directions.
Reviewer 4 Report
2022/10/31 (Sustainability)
Review Comments for Manuscript Number: sustainability-1991752-peer-review-v1
|
Title: |
Numerical Study of Different Engineering Conditions on The Propulsive Performance of the Bionic Jellyfish Robot |
|
Journal: |
Sustainability |
The authors presented a numerical study of the motion of a bionic jellyfish robot. The displacement of the jellyfish robot along the same direction and the surrounding fluid pressure distribution caused by the jellyfish motion under different experimental conditions are discussed considering the environmental conditions. This study showed the importance of frequency on the jellyfish robot’s performance compared to the amplitude. The article is interesting and worthy. I recommend this article after the following:
1. Add a title for the first section (1. Introduction).
2. Enhance your introduction. Less than a half page is presented.
3. Please add the mesh details.
4. Why did you choose COMSOL for such a simulation?
5. Would change the model shape/size/…etc affect the overall performance?
6. Line 97, Does the torque here present the motion or just the outside pressure?
7. I’d like to see a comparison with prior art to check the performance. Add this part to the discussion.
8. Remove section 6 (Patent).
Reviewer 5 Report
This manuscript is very interesting. It is written on the current topic of the movement of robots in a viscous incompressible fluid. The manuscript is well structured and illustrated graphically. The material is presented consistently, the presentation convinces of the correctness of the conclusions. I am sure that the manuscript can be published after the response to the following comments. 1. Why are equations (1)-(3) written for two-dimensional flows? 2. How was the convergence of the numerical integration of the equations of fluid and robot motions studied?
Reviewer 6 Report
1. Line 57: There is no such word as "non-viscous". Instead, it is usually referred to as "inviscid". However, the equations (2) and (3) are actually viscous flow equations (because of the second-order derivatives on the right hand sides), which contradicts with the authors' descriptions.
2. Lines 60, 70, etc.: remove the white spaces before "where".
3. Lines 62 and 63: The definitions of the directions seem do not agree with the equations (4) and (5).
4. How are equations (4)-(6) numerically coupled with the flow equations?
5. Line 88: "as wall conditions without slippage", how is this possible for an inviscid flow?
6. Line 89: "open boundary conditions", please mathematically explain what an "open boundary condition" means.
7. Line 90: "has been automatically meshes in COMSOL", where "meshes" should be "meshed". And, is COMSOL the software used for all the simulations? If so, include this information in, for example, line 50. Otherwise the readers may think you used your own solver for the simulations, while the descriptions on the methods are completely missing.
8. Equations (7) and (8): Add "=" signs to the equations. Also, your simulations are in carried out with dimensions, but you did not mention any thing about the values of mass nor rotational inertial of robot.
9. Figures 3, 4, and 7: What exactly do these "T", "F", and "M" mean in this figures? This information is completely missing.
10. Figure 5: The comparison in this figure is not consistent. The authors claim that they are comparing the frequency effects. However, shouldn't the flow fields be compared at time instants that the robot's becomes identical? Otherwise, the comparison is meaningless, because there are more than one factors that may affect the flow fields.
11. Are the motions of the legs of the robot predefined or calculated by solving the equations? If it is the latter, shouldn't more details be given about the inertial and mass distribution of the robot?
12. This work lacks verification and validation studies.
Overall, the work lacks a lot of information, has contradicting descriptions here and there, and the study is very preliminary. For these reasons, the reviewer would not recommend it for acceptance.